# Knowledge and Attitudes towards Human Papillomavirus Vaccination (HPV) among Healthcare Providers Involved in the Governmental Free HPV Vaccination Program in Shenzhen, Southern China

**DOI:** 10.3390/vaccines11050997

**Published:** 2023-05-18

**Authors:** Danhong Song, Peiyi Liu, Dadong Wu, Fanghui Zhao, Yueyun Wang, Yong Zhang

**Affiliations:** 1National Cancer Center/National Clinical Research Center for Cancer/Cancer Hospital, Chinese Academy of Medical Sciences, Peking Union Medical College, Beijing 100021, China; songdh@student.pumc.edu.cn (D.S.);; 2Affiliated Shenzhen Maternity & Child Healthcare Hospital, Southern Medical University, Shenzhen 518000, China

**Keywords:** HPV vaccine, healthcare provider, knowledge, recommend, influencing factor

## Abstract

No research has been conducted to explore the variables associated with healthcare providers’ (HCPs) knowledge and attitudes toward the human papillomavirus vaccine (HPV) since the vaccine was approved for free use in some Chinese cities. In Shenzhen, southern China, a convenience sample strategy was used to distribute questionnaires to HCPs involved in the government’s HPV vaccination program from Shenzhen. There were 828 questionnaires collected in total, with 770 used in the analysis. The mean HPV and HPV vaccine knowledge score was 12.0 among HCPs involved in the government HPV vaccination program (with a total score of 15). the average scores for HPV and HPV vaccine knowledge varied among different types of medical institutions. District hospitals had the highest mean score of 12.4, while private hospitals ranked fourth with a mean score of 10.9. Multivariate logistic regression results revealed significant disparities in the type of license and after-tax annual income across HCPs (*p* < 0.05). The future education and training for HCPs should focus on private community health centers (CHCs), HCPs whose license type is other than a doctor, and HCPs with low after-tax annual income.

## 1. Introduction

Cervical cancer is the only cancer with a known cause that is both preventable and treatable, yet it is the world’s fourth most common gynecologic cancer, with a significant disease burden in China [1,2]. In 2020, there were 119,300 new cases of cervical cancer and 37,200 deaths, placing the country sixth and seventh in terms of cancer incidence and mortality among Chinese women [2]. In China, the incidence and mortality rates of cervical cancer have increased by varying degrees over the past 20 years [2], and the average age of onset has decreased [3]. Human papillomavirus (HPV) is one of the most common sexually transmitted infections worldwide. High-risk types of HPV are associated with various types of cancer, such as cervical cancer, other anogenital cancers (including penile, vaginal, vulvar, and anal cancers), head and neck cancers (including oral cavity, oropharynx, and larynx cancers), as well as benign warts [4]. The human papillomavirus is responsible for more than 90% of cervical cancer incidences [5]. Clinical studies have demonstrated that HPV vaccines are effective in preventing HPV-related illnesses and reducing the burden of related diseases [6,7,8,9]. Since its introduction in 2006, the HPV vaccine has been gradually implemented in numerous countries. In November 2020, the World Health Organization announced a “Global strategy to accelerate the elimination of cervical cancer as a public health problem”, which included one of the three mid-term strategic objective values for 2030, which aimed for 90% of girls to complete HPV vaccination by the age of 15, achieving primary prevention against HPV infection [10]. As of August 2021, 114 (59%) out of the 194 Members States of the World Health Organization have included HPV vaccination in their national immunization plans [11]. In 2016, China licensed the first HPV vaccinations. However, due to the late adoption of HPV vaccination in China, the government has been gradually implementing a trial program that provides free HPV vaccinations to local governments. At present, the vaccine has not yet been included in the National Immunization Program (NIP). Studies have shown that the cumulative estimated HPV vaccination rate in the female population aged 9–45 in China is only 2.24%, which is relatively low [12]. Awareness and attitudes about the HPV vaccine are not encouraging [13,14,15]. According to a recent school-based nationwide study in China, only 17.1% of adolescents had knowledge of the HPV vaccine [14]. Healthcare provider (HCPs) recommendations are a significant factor that motivates the general population and parents to vaccinate their children against HPV [16,17,18,19]. Additionally, knowledge of HPV-related issues is a crucial predictor of HCP confidence and a willingness to recommend HPV vaccines [20,21,22]. Previous studies have also identified various factors that impact HCPs’ knowledge of HPV and HPV vaccines, including their profession, type of license, age, education level, and job title [23,24]. As Shenzhen is one of the pilot cities for free HPV vaccination, it is essential to assess whether the HCPs participating in the program have the necessary knowledge and willingness to recommend HPV vaccines to school girls.

We conducted a survey of HCPs involved in the free HPV vaccination program for schoolgirls in Shenzhen to describe their current state overall, as well as their knowledge and attitudes toward the HPV vaccine. The survey also aimed to assess the factors that may impact their knowledge and recommended behaviors regarding the vaccine.

## 2. Methods

### 2.1. Study Design and Participants

A cross-sectional study on HPV vaccine knowledge and attitudes among HCPs in Shenzhen was conducted between June 2022 and November 2022. Convenience sampling was used to cover all districts in Shenzhen. The inclusion criteria for the study population included HCPs involved in the governmental free HPV vaccination program in Shenzhen who voluntarily participated in the survey. The exclusion criterion was a refusal to participate in the questionnaire. The survey for this study was collected by sending electronic questionnaires to HCPs who attended training for the free HPV vaccination program in Shenzhen.

### 2.2. Data Collection and Questionnaire

The Chinese online survey application “Questionnaire Star” (https://www.wjx.cn/ (accessed on 1 April 2023)) was utilized to collect the data for this study. Respondents were directed to complete the electronic questionnaire by scanning a QR code or clicking on the link generated by the “Questionnaire Star”. The questionnaire was devised through a collaborative process involving epidemiologists and (HCPs), utilizing the previous literature and extensive discussions. To ensure its validity and effectiveness, the questionnaire was initially tested on a group of 90 HCPs, and adjustments were made to the content and language based on their feedback. The questionnaire was ultimately divided into three sections to cover different aspects of HPV awareness and knowledge. The questionnaire mainly consisted of the following three parts: (1) Socio-demographic data such as age, gender, degree of education, marital status, and income level. (2) Knowledge of HPV and HPV vaccines, such as HPV transmission channels, HPV transmission targets, HPV infection symptoms, and the best time for HPV vaccination. (3) Behavior for recommending HPV vaccination. To calculate the knowledge score, the questionnaire was assigned one point for each correct answer and no points for each incorrect answer. The questionnaire was administered electronically, and it was sent to project training sessions and working groups related to the free HPV vaccination program for schoolgirls in Shenzhen.

### 2.3. Statistical Analysis

HCPs involved in the governmental HPV vaccination program were classified into four groups based on their medical institution and type of employment: Level I/Regional community health center (CHC), Level II CHC, Private CHC, and District Hospitals. The HCPs mainly included obstetricians, gynecologists, general practitioners, public health doctors, and nurses. Chi-square tests were used to compare sociodemographic information between the subgroups, while Kruskal–Wallis H tests and Kruskal–Wallis 1-way ANOVA (k samples) were used to compare knowledge levels. To investigate the characteristics associated with levels of HPV vaccine knowledge, dichotomous logistic regression and multi-variable logistic regression were used(including age, gender, education level, marital status, major, type of license, job title, employment type, after-tax annual income, years of work, and medical institution type). The Mann–Whitney U test was used to compare disparities in the HPV and HPV vaccine knowledge scores among healthcare professionals who recommended the HPV vaccine and those who did not make such recommendations. The Odds Ratio (OR), 95% confidence interval (CI), and *p*-value were determined. Two-tail tests were considered statistically significant if their *p*-values were less than 0.05. SPSS 26.0 (Armonk, NY, USA) was performed for analysis.

## 3. Results

### 3.1. Demographic Characteristics

In total, 828 questionnaires were collected for the study. However, 58 of these questionnaires had missing information or logical errors and were, therefore, excluded from the analysis. The final analysis included 770 records (with a usability rate of 93.0%). Table 1 shows the respondents’ socio-demographic characteristics. The majority of respondents in Level I/Regional CHC (42.3%, *n* = 115), Level II CHC (41.6%, *n* = 128) and Private CHC (22.0%, *n* = 20) were between the ages of 31 and 40. The female respondents’ population was extremely high (94.1%). HCPs from various healthcare institutions differed significantly in age, gender, education level, Major, type of license, job title, employment type, after-tax annual income, and years of work, indicating significant differences (*p* < 0.05) (Table 1).

### 3.2. Knowledge of HPV and HPV Vaccine among Different Types of Medical Institutions

This study found that the mean HPV and HPV vaccine knowledge score was 12.0 among HCPs involved in the government HPV vaccination program (with a total of score 15) and a total knowledge score of 15.0. However, there were several knowledge items that had a correct rate of less than 70%, including: “The body naturally creates high quantities of HPV antibodies to prevent re-infection?” and “HPV can be spread by contact with the skin, oral mucosa, and others” “Who is eligible to receive the HPV vaccine?” This study found that the average scores for HPV and HPV vaccine knowledge varied among different types of medical institutions. District hospitals had the highest mean score of 12.4, while private hospitals ranked fourth with a mean score of 10.9. Further analysis revealed significant differences in the knowledge scores between private community health centers (CHCs) and other medical institution types (adjusted *p* < 0.001), while the other three types of medical institutions did not differ significantly from each other (adjusted *p* = 1.000) (Table 2).

### 3.3. Factors Associated with Knowledge of HPV and the HPV Vaccine among All Participants

Multivariate logistic regression was used to analyze influencing factors regarding HPV and HPV vaccine knowledge levels among the 770 HCP who participated in the government HPV vaccination program. The results revealed significant disparities in the type of license and after-tax annual income across HCPs (*p* < 0.05). Nurses (aOR = 0.26, 95% CI: 0.18–0.38) and other HCPs (aOR = 0.25, 95% CI: 0.12–0.51) were lower than physicians. This study found that HCPs with higher after-tax annual incomes had higher HPV and HPV vaccine knowledge scores. Using an after-tax annual income of 100,000 RMB as a reference, HCPs with incomes of 100,000–200,000 RMB (aOR = 1.61, 95% CI: 1.07–2.42), 200,000–300,000 RMB (aOR = 2.16, 95% CI: 1.24–3.76), and >300,000 RMB (aOR = 2.39, 95% CI: 0.89–6.42) had higher adjusted odds ratios for higher knowledge scores (Table 3).

### 3.4. HPV Vaccination Recommendation Behavior

Out of 770 participants, a total of 729 (94.7%) reported recommending the HPV vaccine to others. This included 259 (94.9%) in Level I/Regional CHCs, 292 (94.8%) in Level II CHCs, 86 (94.5%) in Private CHCs, and 92 (93.9%) in district hospitals. However, some respondents reported a reluctance to recommend the HPV vaccine to others, citing several reasons. The main three reasons for this included: “Vaccine promotion is not my responsibility,” “Fear of trouble caused by recommending self-pay vaccines to service recipients” and “Uncertainty of the HPV vaccination process” (Table 4).

In addition, the Mann–Whitney U test was used to analyze the differences in HPV and HPV vaccine knowledge scores between healthcare professionals who recommended the HPV vaccine and those who had not recommended it. The results showed that the distribution of HPV and HPV vaccine knowledge scores between the two groups of healthcare professionals was inconsistent. The average knowledge score of healthcare professionals who recommended the HPV vaccine was 12.09 ± 2.02, while that of healthcare professionals who did not recommend the HPV vaccine was 10.76 ± 2.67. The average rank of knowledge scores for healthcare professionals who had recommended the HPV vaccine was 392.15, while that of healthcare professionals who had not recommended the HPV vaccine was 267.30. The Mann–Whitney U test results indicated that there was a statistically significant difference in HPV and HPV vaccine knowledge scores between healthcare professionals who had recommended the HPV vaccine and those who had not recommended it (U = 10,098.500, *p* < 0.001).

## 4. Discussion

In the context of the gradual introduction of the free HPV vaccination program in Chinese cities since 2020, there has been a need for evaluation studies on HCPs associated with this program. This study aimed to assess the level of HPV and HPV vaccine knowledge among HCPs and examine differences between the knowledge levels among HCPs in various types of medical institutions. In addition, we explored the factors that influenced the level of HPV and HPV vaccine knowledge among HCPs and identified the reasons for not recommending the HPV vaccine. The findings of this study provided valuable insights for improving the overall knowledge of HCPs involved in the program and promoting the quality of program implementation in the region.

This study found that HCPs in private CHCs had lower knowledge levels about HPV and the HPV vaccine compared to HCPs in the other three public medical institutions. The difference in knowledge levels between the four medical institutions was significant. However, the recommendation behaviors of HCPs across the four medical institutions were consistent. This study also identified that HCPs’ knowledge of the HPV vaccine was influenced by their type of license and after-tax annual income. Additionally, the most common reason given for not recommending the HPV vaccine to their clients was that “vaccine promotion is not my responsibility”. These findings highlight the importance of targeted education and training programs for HCPs, particularly those working in private CHCs, to improve their knowledge and recommendation behavior around the HPV vaccine. It also suggests the need for greater emphasis on the importance of HPV vaccination and the role of HCPs in promoting it to their patients. The survey revealed significant differences in the sociological characteristics of HCPs involved in the government HPV vaccination program across the four types of medical institutions, except for marital status. HCPs working in private CHCs had a lower age distribution, a higher percentage of male HCPs, fewer preventive medicine majors, fewer doctors, more nurses, no formal staffing, lower education levels, lower job titles, and a lower after-tax annual income. Additionally, their years of work were primarily distributed between more than 21 and less than 5 years. These findings suggest that private CHCs may face challenges in terms of their talent pool, as their HCPs tend to have lower qualifications and experience when compared to those in other medical institutions. Addressing these disparities may require targeted efforts to improve the recruitment and retention of qualified healthcare professionals, as well as targeted education and training programs to improve their knowledge and skills.

Our findings are consistent with the findings in the Pearl River Delta region [25,26]. Owing to Shenzhen’s developed economy and recent emphasis on the construction of primary medical institutions, as well as the good stability and high credibility of public medical institutions, studies have noted a trend in staff from private CHCs migrating to public health institutions after gaining several years of training and experience. This has resulted in lower talent levels in private CHCs compared to public medical institutions. Addressing this issue may require efforts to improve the recruitment and retention of qualified healthcare professionals in private CHCs, as well as targeted education and training programs to improve the knowledge and skills of their HCPs.

This study found that the mean (SD) HPV and HPV vaccine knowledge score among participants in Shenzhen was 12.02 (2.08) out of 15 [27,28,29]; this suggests that there is room for improvement in HCPs’ knowledge and understanding of HPV and the HPV vaccine in Southern China. This could be due to China’s late approval of the HPV vaccine and the absence of NIP implementation. More than 96.0% correctly identified “regular cervical cancer screening is required after HPV vaccination” based on the correct rate of each knowledge item. “Regular cervical cancer screening for women is an important preventive measure”. This may be because cervical cancer screening has been a national public health program since 2009. However, only 52.2% were aware that “the body’s natural infection with HPV has a low level of resistance that is insufficient to fight off another virus attack”. Less than 65% were aware that “HPV could be transmitted through contact with the skin, oral mucosa, and other body fluids” and that “HPV vaccination is available for both men and women”. This suggests that while most HCPs are familiar with HPV and the HPV vaccine, they lack in-depth knowledge of HPV infection pathways and modes of transmission. Future educational programs or training courses should take care to explain these items of knowledge. Studies in countries with HPV vaccination programs have also revealed that medical personnel’s knowledge of HPV and HPV vaccines is frequently incomplete, with the potential to spread misinformation [27,30,31]. Furthermore, studies in countries with established HPV vaccination programs also revealed that medical personnel’s knowledge of HPV and HPV vaccines was frequently incomplete, which could lead to the spread of misinformation. Therefore, it is important for educational programs to be based on accurate and up-to-date information and to be regularly updated as new research emerges. In doing so, HCPs can be better equipped to provide accurate information and recommendations to their patients, leading to improved vaccination rates and reduced rates of HPV-related diseases.

A comparison of the four types of medical institutions revealed that there was no difference in the level of knowledge between the three types of public medical institutions but there was a significant difference between private CHC and the other three types of public medical institutions. This disparity could be attributed to the lower overall quality of medical staff in private CHCs compared to public medical institutions. Therefore, it is critical to increase HCPs’ knowledge of HPV and HPV vaccines, improve access to HPV and HPV vaccine information in private CHCs, and strengthen HPV education for HCPs in CHCs. These efforts are necessary to ensure that HCPs are equipped with the information and skills required to offer effective HPV prevention and treatment to patients. Overall, targeted education and training programs, as well as increased access to information and resources, are essential for improving HCPs’ knowledge and understanding of HPV and the HPV vaccine. In doing so, we can strive towards reducing the incidence of HPV-related diseases and improving the overall health outcomes of patients in Southern China. Consistent with previous studies [24,32], our study found that the type of license and income level were significant factors influencing knowledge levels of HPV and HPV vaccines among HCPs in Southern China. Specifically, doctors exhibited a higher level of knowledge compared to nurses and other HCPs, and individuals with a higher after-tax annual income had a higher level of knowledge. Other studies also highlighted the importance of effective communication between HCPs and females in raising awareness and acceptance of HPV vaccines [33]. A previous study reported that 78% of female participants expressed interest in receiving more information about HPV from their doctors [34]. Furthermore, another study found that approximately 60% of women who were willing to vaccinate their children against HPV cited their doctor’s advice as a critical factor in their decision [35]. In addition, the doctor’s advice played an important role in increasing parental willingness to vaccinate their children against HPV. According to the results of this study, 94.7% of the participants recommended the HPV vaccine to their service recipients, and 5.3% did not recommend the HPV vaccine mainly because “vaccine promotion is not my responsibility” and “fear of trouble caused by recommending self-pay vaccines to service recipients”. Additionally, HCP with a higher level of knowledge about HPV and HPV vaccines are more likely to recommend the vaccine to their patients. In order to further improve public awareness and the use of the HPV vaccine in the population, training for HCPs involved in the government’s HPV vaccination program should not only focus on those with lower levels of knowledge, but also on strengthening HCPs’ awareness of publicity and education. This could help to ensure that HCPs are equipped with accurate and up-to-date information about HPV and the HPV vaccine, further enhancing their willingness to recommend the HPV vaccine and increasing their ability to effectively communicate this information with their patients. In addition to this, schools and the general public could also be given information and education about HPV and the vaccine through HCPs and the media. In doing so, we can ensure that people are better informed about HPV-related knowledge and the implications of the HPV vaccine, as well as clear up any misconceptions about HCPs recommending the HPV vaccine. This can ultimately lead to an increase in the rate of HPV vaccination and a reduction in rates of HPV-related diseases in China.

## 5. Strengths and Limitations

The present study has several strengths that are worth noting. Firstly, it is the first survey conducted in China that specifically explores HPV and HPV vaccine knowledge and attitudes among HCPs involved in the government’s HPV vaccination program in Shenzhen. This contributes to the existing literature and fills an important knowledge gap in this field. Secondly, our sample size was relatively large, with 770 HCPs participating in this study. This allowed for a more comprehensive analysis of the knowledge and attitudes of HCPs towards HPV and HPV vaccines. Lastly, we conducted a detailed analysis of the factors influencing HCPs’ knowledge levels and recommended behaviors, which could provide valuable insights for improving HCPs’ education and training programs in relation to HPV vaccination.

Several limitations of our study should be acknowledged. Firstly, the questionnaire used in this study was only implemented in Shenzhen and was developed based on the national setting of China. Secondly, the findings are restricted to the specific data obtained from Shenzhen. Therefore, caution should be taken when applying these results to other regions where legislative and health-related implementations differ from those in Shenzhen. However, the results are still valuable for promoting the implementation of the free HPV vaccination program in Shenzhen. Lastly, as the study adopted a convenience sample instead of a probability sample, there may be variation in the level of access among participants. Therefore, future studies are recommended to use random sampling and perform rigorous analyses.

## 6. Conclusions

This study provides an overview of Shenzhen, southern China, where HCPs exhibit a higher level of knowledge and recommended behaviors for the HPV vaccine. The knowledge that HCPs have on HPV infection routes and modes of transmission, as well as some of the reasons for non-recommended behaviors, needs to be improved. The significantly lower scores of HCPs in private CHCs compared to public medical institutions, as well as the factors influencing knowledge levels, indicate that future education and training for HCPs should focus on private CHCs, HCPs whose license type is other than a doctor, and HCPs with a low after-tax annual income.

## Figures and Tables

**Table 1 vaccines-11-00997-t001:** Demographic haracteristics of HCPs involved in the governmental HPV vaccination program (*n* = 770).

Variables	Total		Level I/Regional CHC * (%)		Level II CHC * (%)		Private CHC * (%)		District Hospitals (%)		X^2^	*p*
Age ^a^	770		273		308		91		98			
≤30	161	21.00%	61	22.40%	56	18.20%	32	35.20%	12	12.40%	51.542	<0.001
31~40	299	38.90%	115	42.30%	128	41.60%	20	22.00%	36	37.10%		
41~50	246	32.00%	83	30.50%	105	34.10%	20	22.00%	38	39.20%		
>50	62	8.10%	13	4.80%	19	6.20%	19	20.90%	11	11.30%		
Gender												
Male	56	7.30%	16	5.90%	23	7.50%	13	14.30%	4	4.10%	8.941	0.03
Female	714	92.70%	257	94.10%	285	92.50%	78	85.70%	94	95.90%		
Education level												
Junior High School/High School/Vocational High School/Junior College	21	2.70%	6	2.20%	5	1.60%	9	9.90%	1	1.00%	31.293	<0.001
College/University	688	89.40%	250	91.60%	273	88.60%	81	89.00%	84	85.70%		
Master’s degree or above	61	7.90%	17	6.20%	30	9.70%	1	1.10%	13	13.30%		
Marital Status												
Unmarried	126	16.40%	52	19.00%	42	13.60%	20	22.00%	12	12.20%	6.801	0.34
Married	607	78.80%	208	76.20%	252	81.80%	67	73.60%	80	81.60%		
Divorced/widowed	37	4.80%	13	4.80%	14	4.50%	4	4.40%	6	6.10%		
Major												
Clinical Medicine	409	53.10%	153	56.00%	163	52.90%	46	50.50%	47	48.00%	26.661	0.002
Preventive Medicine	48	6.20%	14	5.10%	17	5.50%	2	2.20%	15	15.30%		
Nursing	221	28.70%	84	30.80%	80	26.00%	31	34.10%	26	26.50%		
Other HCP	92	11.90%	22	8.10%	48	15.60%	12	13.20%	10	10.20%		
Type of license												
Doctors	511	66.40%	181	66.30%	219	71.10%	49	53.80%	62	63.30%	39.891	<0.001
Nurse	218	28.30%	83	30.40%	83	26.90%	30	33.00%	22	22.40%		
Other HCP	41	5.30%	9	3.30%	6	1.90%	12	13.20%	14	14.30%		
Job title												
Lower than primary	27	3.50%	6	2.20%	5	1.60%	9	9.90%	7	7.10%	105.31	<0.001
Primary	204	26.50%	74	27.10%	65	21.10%	47	51.60%	18	18.40%		
Intermediate	437	56.80%	164	60.10%	207	67.20%	26	28.60%	40	40.80%		
Deputy senior/Senior	102	13.20%	29	10.60%	31	10.10%	9	9.90%	33	33.70%		
Employment Type ^b^												
temporary employment	247	32.20%	94	34.60%	102	33.20%	20	22.20%	31	32.00%	83.807	<0.001
Contract Employment	325	42.40%	113	41.50%	127	41.40%	62	68.90%	23	23.70%		
Formal staffing	175	22.80%	58	21.30%	75	24.40%	0	0.00%	42	43.30%		
Retirement and re-employment	19	2.50%	7	2.60%	3	1.00%	8	8.90%	1	1.00%		
After-tax annual income												
<100,000 RMB	158	20.50%	54	19.80%	40	13.00%	48	52.70%	16	16.30%	89.923	<0.001
100,000~200,000 RMB	438	56.90%	150	54.90%	201	65.30%	40	44.00%	47	48.00%		
200,000~300,000 RMB	143	18.60%	58	21.20%	57	18.50%	2	2.20%	26	26.50%		
>300,000 RMB	31	4.00%	11	4.00%	10	3.20%	1	1.10%	9	9.20%		
Years of work ^c^												
<5 years	151	19.70%	53	19.50%	60	19.50%	26	28.90%	12	12.20%	31.078	0.002
6–10 years	138	18.00%	63	23.20%	47	15.30%	16	17.80%	12	12.20%		
11–15 years	127	16.60%	44	16.20%	56	18.20%	13	14.40%	14	14.30%		
16–20 years	136	17.70%	44	16.20%	66	21.50%	8	8.90%	18	18.40%		
>21 years	215	28.00%	68	25.00%	78	25.40%	27	30.00%	42	42.90%		

* CHC: community health center. ^a^ Two-missing value in HCPs involved in the governmental HPV vaccination program age. ^b^ Four missing values in HCPs involved in the governmental HPV vaccination program Employment Type. ^c^ Three missing value in HCPs involved in the governmental HPV vaccination program in years of work.

**Table 2 vaccines-11-00997-t002:** Knowledge of HPV and HPV vaccines among different medical institution types (*n* = 770).

Question/Correct	Total		Level I/Regional CHC		Level II CHC		Private CHC		District Hospitals	
	N	%	N	%	N	%	N	%	N	%
HPV-related questions										
The majority of HPV infections in people do not cause any symptoms?	626	81.3%	224	82.1%	256	83.1%	63	69.2%	83	84.7%
Autoimmune therapy can cure the majority of HPV infections?	541	70.3%	192	70.3%	222	72.1%	46	50.5%	81	82.7%
The body naturally creates high quantities of HPV antibodies to prevent re-infection?	402	52.2%	134	49.1%	169	54.9%	57	62.6%	42	42.9%
The patient or the individual who has the virus are not the source of HPV transmission?	559	72.6%	209	76.6%	225	73.1%	59	64.8%	66	67.3%
HPV can be spread through sexual contact?	731	94.9%	263	96.3%	290	94.2%	84	92.3%	94	95.9%
During childbirth, a mother’s genital tract HPV infection may pass to the baby?	549	71.3%	189	69.2%	233	75.6%	63	69.2%	64	65.3%
HPV can be spread by contact with the skin, oral mucosa, and others?	483	62.7%	175	64.1%	193	62.7%	42	46.2%	73	74.5%
Only women can contract HPV?	706	91.7%	259	94.9%	277	89.9%	75	82.4%	95	96.9%
Women exclusively contract HPV in their cervix?	672	87.3%	243	89.0%	268	87.0%	71	78.0%	90	91.8%
Regular cervical cancer screening for women is an important measure to prevent the disease?	739	96.0%	260	95.2%	296	96.1%	86	94.5%	97	99.0%
HPV vaccine-related questions										
Who is eligible to receive the HPV vaccine?	500	64.9%	186	68.1%	199	64.6%	47	51.6%	68	69.4%
Who is the front-runner for the human papillomavirus (HPV) vaccine?	580	75.3%	208	76.2%	236	76.6%	59	64.8%	77	78.6%
What time of year is ideal for receiving the human papillomavirus (HPV) vaccine?	713	92.6%	255	93.4%	292	94.8%	75	82.4%	91	92.9%
The human papillomavirus (HPV) vaccine guards against the virus?	708	91.9%	250	91.6%	290	94.2%	75	82.4%	93	94.9%
After receiving an HPV vaccine, routine cervical cancer screenings are no longer necessary?	748	97.1%	267	97.8%	296	96.1%	89	97.8%	96	98.0%
Overall mean score	12.0	80.0%	12.1	80.7%	12.1	80.7%	10.9	72.7%	12.4	82.3%

Kruskal–Wallis H: H = 28.441, *p* < 0.001. Kruskal–Wallis one-way ANOVA (k samples): Private CHC-Level I/Regional CHC (adjsted *p* < 0.001); Private CHC-Level II CHC (adjsted *p* < 0.001); Private CHC-District Hospitals (adjsted *p* < 0.001); Level I/Regional CHC-Level II CHC (adjsted *p* = 1.000); Level I/Regional CHC-District Hospitals (adjsted *p* = 1.000); Level II CHC-District Hospitals (adjsted *p* = 1.000).

**Table 3 vaccines-11-00997-t003:** Factors associated with the knowledge of HPV and the HPV vaccine among all participants (*n* = 770).

Variables	Average Score	Score < 12	Score ≥ 12	Uni-Variate Logistic Regression		Multi-Variate Logistic Regression	
				OR (95% CI)	*p*	Aor (95% CI)	*p*
Age ^a^	12.02 ± 2.08				0.060		0.127
≤30	11.68 ± 2.17	63	98	1.00		1.00	
31~40	12.01 ± 2.11	100	180	1.28 (0.86–1.90)	0.224	-	0.714
41~50	12.33 ± 1.87	66	199	1.75 (1.15–2.68)	0.009	-	0.429
>50	11.85 ± 2.28	23	39	1.09 (0.60–2.00)	0.780	-	0.027
Gender					0.289		
Male	11.84 ± 2.05	22	34	1.00		*	*
Female	12.04 ± 2.09	231	483	1.35 (0.77–2.37)	0.289	*	*
Education level					<0.001		0.230
Junior High School/High School/Vocational High School/Junior College	10.33 ± 2.24	15	6	1.00		1.00	
College/University	12.01 ± 2.09	226	462	5.11 (1.96–13.35)	0.001	-	0.465
Master’s degree or above	12.79 ± 1.56	12	49	10.21 (3.27–31.85)	<0.001	-	0.775
Marital Status					0.879		
Unmarried	11.85 ± 2.31	43	83	1.00		*	*
Married	12.04 ± 2.01	199	408	1.06 (0.71–1.60)	0.771	*	*
Divorced/widowed	12.30 ± 2.42	11	26	1.23 (0.55–2.71)	0.618	*	*
Major					<0.001		0.577
Clinical Medicine	12.54 ± 1.76	95	314	1.00		1.00	
Preventive Medicine	12.71 ± 1.58	8	40	1.51 (0.68–3.34)	0.306	-	0.309
Nursing	10.83 ± 2.32	120	101	0.26 (0.18–0.36)	<0.001	-	0.358
Other HCP	12.22 ± 1.90	30	62	0.63 (0.38–1.02)	0.062	-	0.586
Type of license					<0.001		<0.001
Doctors	12.61 ± 1.72	110	401	1.00		1.00	
Nurse	10.82 ± 2.31	120	98	0.22 (0.16–0.32)	<0.001	0.26 (0.18–0.38)	<0.001
Other HCP	11.12 ± 1.98	23	18	0.22 (0.11–0.41)	<0.001	0.25 (0.12–0.51)	<0.001
Job Title					0.001		0.803
Lower than primary	10.89 ± 2.26	15	12	1.00		1.00	
Primary	11.44 ± 2.21	84	120	1.79 (0.80–4.01)	0.160	-	0.838
Intermediate	12.21 ± 2.00	129	308	2.98 (1.36–6.55)	0.006	-	0.630
Deputy senior/Senior	12.71 ± 1.76	25	77	3.85 (1.59–9.31)	0.003	-	0.521
Employment Type ^b^					0.007		0.697
Temporary employment	11.90 ± 2.17	85	162	1.00		1.00	
Contract Employment	11.70 ± 2.09	122	203	0.87 (0.62–1.23)	0.441	-	0.937
Formal staffing	12.82 ± 1.66	39	136	1.83 (1.18–2.85)	0.007	-	0.407
Retirement and re-employment	11.74 ± 2.68	6	13	1.14 (0.42–3.10)	0.802	-	0.380
After-tax annual income					<0.001		0.029
<100,000 RMB	10.88 ± 2.28	81	77	1.00		1.00	
100,000~200,000 RMB	12.09 ± 1.99	136	302	2.34 (1.61–3.39)	<0.001	1.61 (1.07–2.42)	0.023
200,000~300,000 RMB	12.80 ± 1.62	30	113	3.96 (2.38–6.59)	<0.001	2.16 (1.24–3.76)	0.007
>300,000 RMB	13.35 ± 1.56	6	25	4.38 (1.71–11.27)	0.002	2.39 (0.89–6.42)	0.083
Years of work ^c^					0.557		
<5 years	12.09 ± 2.08	48	103	1.00			
6–10 years	11.75 ± 2.14	53	85	0.75 (0.46–1.21)	0.239	*	*
11–15 years	11.92 ± 1.94	42	85	0.94 (0.57–1.56)	0.820	*	*
16–20 years	12.01 ± 2.36	43	93	1.01 (0.61–1.66)	0.975	*	*
>21 years	12.24 ± 1.94	64	151	1.10 (0.70–1.73)	0.680	*	*
Medical institution Type					0.002		0.187
Level I/Regional CHC	12.14 ± 1.81	83	190	1.00		1.00	
Level II CHC	12.15 ± 2.15	92	216	1.03 (0.72–1.46)	0.889	-	0.667
Private CHC	10.89 ± 2.41	46	45	0.43 (0.26–0.69)	0.001	-	0.038
District Hospitals	12.35 ± 1.94	32	66	0.90 (0.55–1.48)	0.680	-	0.769

^a^ Two-missing value in HCPs involved in the governmental HPV vaccination program age. ^b^ four missing value in HCPs involved in the governmental HPV vaccination program Employment Type. ^c^ Three missing value in HCPs involved in the governmental HPV vaccination program in years of work * Not included in multifactorial analysis—Variables not included in the equation.

**Table 4 vaccines-11-00997-t004:** HPV Vaccine recommendation behavior (*n* = 770).

Variables	Total		Level I/Regional CHC		Level II CHC		Private CHC		District Hospitals	
	N	%	N	%	N	%	N	%	N	%
Have you ever advised a client to receive an HPV vaccine?	770		273		308		91		98	
Yes	729	94.70%	259	94.90%	292	94.80%	86	94.50%	92	93.90%
No	41	5.30%	14	5.10%	16	5.20%	5	5.50%	6	6.10%
The reason you did not recommend HPV vaccination to your clients (multiple choice)										
Workplaces are not allowed to recommend self-funded vaccines to clients	5	12.20%	4	28.60%	1	6.30%	5	100.00%	0	0.00%
Uncertain who the HPV vaccine is intended for.	6	14.60%	1	7.10%	4	25.00%	0	0.00%	1	16.70%
Uncertain of the HPV vaccination process	12	29.30%	2	14.30%	7	43.80%	2	40.00%	1	16.70%
Uncertain about safety of HPV vaccine	8	19.50%	3	21.40%	3	18.80%	1	20.00%	1	16.70%
Uncertain effectiveness of HPV vaccine	7	17.10%	2	14.30%	3	18.80%	0	0.00%	2	33.30%
Fear of trouble caused by recommending self-pay vaccines to service recipients	14	34.10%	4	28.60%	7	43.80%	2	40.00%	1	16.70%
Vaccine promotion is not my responsibility	15	36.60%	4	28.60%	8	50.00%	1	20.00%	2	33.30%
Other	6	14.60%	4	28.60%	0	0.00%	1	20.00%	1	16.70%

## Data Availability

The data that support the findings of this study are available from the corresponding author.

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
