# Peer review of "Knowledge and Attitudes towards Human Papillomavirus Vaccination (HPV) among Healthcare Providers Involved in the Governmental Free HPV Vaccination Program in Shenzhen, Southern China"

_vaccines, 2023, doi:10.3390/vaccines11050997_

Round 1

Reviewer 1 Report

Manuscript (ID: vaccines-2359278) presents results of a survey of HCPs involved in the free HPV vaccination program for schoolgirls in Shenzhen, China, to assess the factors that may impact their knowledge and attitudes towards the HPV vaccine. However, there are some inaccuracies in this manuscript and major revision is necessary. Comments:      

  • Lines 27-62: Section Introduction presents in detail the existing knowledge on the topic of this study and gives an overview of the status of HPV vaccination in China, citing the relevant and most current literature.    

  • Lines 28-30: In addition to reference No. 1, which is cited in this sentence, reference No. 2 (from current list of References) in this sentence should be added. Rationale: In the cited reference No. 1, data on `a significant disease burden in China' for cervical cancer are not explicitly stated. 

  • Lines 63-66: The objectives of this study are clearly defined.   

  • Lines 72-74: Cite the appropriate reference for the statement `previously observed rate of 74% (with a 95% confidence interval of 63% to 86%).`. 

  • Line 74: Add paragraphs that will present information on 
    • `Study population', and 
    • `Study sample', with inclusion criteria and exclusion criteria, 
    • `Participation rate`, 
    • `Response rate`.  

  • Lines 99-106: Clarify this text, remove duplicate text and specify the purpose for which the given variables are listed.   

  • Lines 106-107: State which condition the observed variables had to meet in order to be entered into the multivariable logistic regression model. 

  • Line 106-107: State the results of the collinearity test between the observed variables. 

  • Lines 106-107: In Table 3 (on Lines 164-168) in this manuscript data for `aOR', that is for adjusted odds ratios, are shown. Indicate in subsection Statistical analysis for which variables the adjustment was carried out. 

  • Lines 115-116: Correct this statement and specify it in such a way as to state that `The majority of respondents (42.3%, n=115) were between the ages of 31 and 40 among respondents from Level I/Regional CHC.' . 

  • Lines 116-117: Correct this sentence to state that it is about the representation (share) of women in this study sample, and not ``The female population was extremely high (94.1%)'' per se.  

  • Line 154: Check if an explanation is given below the table for marking with `*` (for: Level I/Regional CHC* (%); Level II CHC*(%); Private CHC*(%)) in the subtitle of Table 1. Correct this.  

  • Lines 158, 164 and 169: Insert `(n=770).` in titles of Tables 2-4.  

  • Lines 159-163: Clarify the use of the terms `P<0.001' and `adjusted P<0.001)'.     

  • Line 168: List all abbreviations presented in Table 3. E.g. `aOR', with a list of all variables for which adjustment was carried out, etc.  

  • Lines 171-178: Instead of this paragraph (which is a repetition of what was already mentioned above), enter a paragraph in which the most important results of this study will be highlighted.   

  • Line 179-275: In this text in the Discussion section, the comparison of the results of this study with the results of similar research in the world is presented in a good manner.  

  • Line 275: Add a new paragraph in which the Strengths and Limitations of this study will be discussed in detail.  

  • Line 276: Section Conclusions correctly summarizes the most important results of this study. 

The quality of English language is acceptable.   

Author Response

Point1: Lines 28-30: In addition to reference No. 1, which is cited in this sentence, reference No.

2 (from current list of References) in this sentence should be added. Rationale: In the cited

reference No. 1, data on `a significant disease burden in China' for cervical cancer are not

explicitly stated.

Response 1:Line 31—reference No. 2 be added.

Point2: Line 74: Add paragraphs that will present information on

 `Study population', and

 `Study sample', with inclusion criteria and exclusion criteria,

 `Participation rate`,

 `Response rate`.

Response 2: Lines80-83 Add study sample

“The inclusion criteria for the study population were healthcare providers involved in the

governmental free HPV vaccination program in Shenzhen who voluntarily participated in the

survey. The exclusion criterion was refusal to participate in the questionnaire. The survey for this

study was collected by sending electronic questionnaires to healthcare providers who attended the

training for the free HPV vaccination program in Shenzhen.”

Lines127-129 Descriptions response rate

In total, 828 questionnaires were collected for the study. However, 58 of these questionnaires had

missing information or logical errors, and were therefore excluded from the analysis. The final

analysis included 770 records (with a validity rate of 93.0%).

Point 3:Lines 115-116: Correct this statement and specify it in such a way as to state that `The

majority of respondents (42.3%, n=115) were between the ages of 31 and 40 among respondents

from Level I/Regional CHC

Response 3:Lines 131-132 The majority of respondents in Level I/Regional CHC(42.3%,

n=115),Level II CHC(41.6%, n=128) and Private CHC(22.0%, n=20) were between the ages of 31

and 40.

Point 4:Lines 116-117: Correct this sentence to state that it is about the representation (share) of

women in this study sample, and not ``The female population was extremely high (94.1%)'' per

se.

Response 4: Lines 132-133 The female respondents' population was extremely high (94.1%).

Point 5: Line 154: Check if an explanation is given below the table for marking with `*` (for:

Level I/Regional CHC* (%); Level II CHC*(%); Private CHC*(%)) in the subtitle of Table 1.

Correct this.

Response 5: Line 153 *CHC: community health center.

Point 6:Line 158, 164, and 169: Insert "(n=4)" in the title of Table 770-2.

Response 6:insert line 152、192 and 199(n=770)

Point 7::Lines 171-178: Replace this paragraph (which is a repetition of the content already

mentioned above) and instead input a paragraph that will highlight the most important findings of

this study.

Response 7:line 207-215 Instead of the paragraph :"In the context of the gradual introduction of

the free HPV vaccination program in Chinese cities since 2020, there is a need for evaluation

studies on HCP associated with this program. This study aimed to assess the level of HPV and

HPV vaccine knowledge among HCP and examine differences in knowledge levels among HCP

in various types of medical institutions. In addition, we explored the factors that influence the

level of HPV and HPV vaccine knowledge among HCP and identified reasons for not

recommending the HPV vaccine. The findings of this study provide valuable insights for

improving the overall knowledge of HCPs involved in the program and promoting the quality of

program implementation in the region. 。

Point 8:Line 275: Add a new paragraph to discuss in detail the strengths and limitations of this

study.

Response 8:Add a new paragraph from line 321 to 341.

Strengths and limitations

The present study has several strengths worth noting. Firstly, it is the first survey conducted in

China that specifically explores HPV and HPV vaccine knowledge and attitudes among HCPs

involved in the government's HPV vaccination program in Shenzhen. This contributes to the

existing literature and fills an important knowledge gap in this field. Secondly, our sample size

was relatively large, with 770 HCPs participating in the study. This allowed for a more

comprehensive and representative analysis of the knowledge and attitudes of HCPs towards HPV

and HPV vaccines. Lastly, we conducted a detailed analysis of the factors influencing HCPs'

knowledge levels and recommended behaviors, which can provide valuable insights for improving

HCPs' education and training programs related to HPV vaccination.

Several limitations of our study should be acknowledged. Firstly, the questionnaire used in the

study was only implemented in Shenzhen and was developed based on the national setting of

China. Secondly, the findings are restricted to the specific data obtained from Shenzhen. Therefore,

caution should be taken when applying the results to other regions where legislative and

health-related implementations differ from those in Shenzhen. However, the results are still

valuable for promoting the implementation of the free HPV vaccination program in Shenzhen.

Lastly, as the study adopted a convenience sample instead of a probability sample, there may be

variation in the level of access among participants. Therefore, future studies are recommended to

use random sampling and perform rigorous analyses.

Reviewer 2 Report

In this manuscript the authors report survey results of a convenience sample of health care providers about HPV/vaccination in Shenzhen, China. The manuscript is primarily descriptive, describing how the providers feel/think about HPV and factors that affect their thoughts.

Abstract:

·        The abstract states first that 818 HCPs were contacted but then 828 questionnaires were returned. Why were more questionnaires returned than the number of people contacted? Since this is a convenience sample, I would report only the number that did respond, the number of surveys that were used in the analysis, and the number that you were aiming for at a minimum. (Also, since the authors never report the 818/828 values in the methods section, this is information that can also be left out of the abstract.)

·        When discussing the knowledge score, you should refer to the total score of 15 as the total possible score. As written (here and throughout the manuscript), it sounds like people had a total score of 15 rather than that there was a possibility of getting up to 15. Stating it this way (as a “possible” total score) will make clearer the comparison between means that are presented in the abstract.

Introduction:

·        Expectations about what factors may be important in explaining thoughts/behaviors with respect to HPV and its vaccine should be briefly discussed. What does prior research on other vaccines/diseases in general indicate are important socio-demographic variables? The authors end up including several variables as explanatory factors but they are presented as a “fishing expedition” rather than presenting a rationale why they all should be included. The introduction should include a brief discussion of what background factors have been deemed important for other research and are therefore included here.

Methods:

·        The authors state (line 71) that a target sample size of 700 was determined as a way to get accurate estimation. However, this is a convenience sample. Without using probability sampling, you have no way of determining the degree to which the sample is “accurate” or representative. The lines from 71-74 do not make sense with this kind of sampling.

·        More information on how the people were contacted for the survey is needed. If the questionnaire was sent to various places, how did they know that they would reach at least 700? The whole sampling and administrative procedure needs more explanation.

Results:

·        How was a “validity rate” determined? What is meant by this? (line 114)

·        In running the chi-square analyses (table 1), were there any issues with small expected cell sizes? Some of the cell frequency values are quite small which likely leads to problems with the chi-square test.

·        Table 2—I would indicate i the table title that this is the N and % of those getting the answers correct. This is not clear as it is currently formatted.

·        I’m curious as to why the authors did not look at the socio-demographic factors as possible predictors for whether an HCP has recommended the HPV vaccine. Are there too few people in the “no” category for meaningful analysis? (This then also highlights the point I make above about the potential problem with chi-square analysis.)

Discussion:

·        It is interesting to see that in China, it is the private practices that have the lowest quality and knowledge. This is very different from the U.S. context where the private practices would have the highest quality, knowledge, and pay.

Extensive editing is needed

Author Response

Point 1:Abstrac

1.The abstract states first that 818 HCPs were contacted but then 828 questionnaires were returned. Why were more questionnaires returned than the number of people contacted? Since this is a convenience sample, I would report only the number that did respond, the number of surveys that were used in the analysis, and the number that you were aiming for at a minimum. (Also, since the authors never report the 818/828 values in the methods section, this is information that can also be left out of the abstract.)

2.When discussing the knowledge score, you should refer to the total score of 15 as the total possible score. As written (here and throughout the manuscript), it sounds like people had a total score of 15 rather than that there was a possibility of getting up to 15. Stating it this way (as a “possible” total score) will make clearer the comparison between means that are presented in the abstract.

Respons 1:

  1. This was my mistake. In fact, a total of 828 questionnaires were received, and the content has been updated accordingly.Line14-17

“In Shenzhen, southern China, a convenience sample strategy was used to distribute questionnaires to HCPs involved in the government's HPV vaccination program from Shenzhen. There were 828 questionnaires collected in total, with 770 being used in the analysis.”

  1. The expression has been modified to "total score 15".line 17-18

“The mean HPV and HPV vaccine knowledge score was 12.0 among HCPs involved in the government HPV vaccination program(total score 15)”

Point 2: introduction

Expectations about what factors may be important in explaining thoughts/behaviors with respect to HPV and its vaccine should be briefly discussed. What does prior research on other vaccines/diseases in general indicate are important socio-demographic variables? The authors end up including several variables as explanatory factors but they are presented as a “fishing expedition” rather than presenting a rationale why they all should be included. The introduction should include a brief discussion of what background factors have been deemed important for other research and are therefore included here.

Respons 2: Relevant explanations have been added in the introduction section.(line 61-63)

“Previous studies have also identified various factors that impact HCPs' knowledge of HPV and HPV vaccines, including their profession, type of license, age, education level, and job title[23,24].”

Point 3: Methods

“1. The authors state (line 71) that a target sample size of 700 was determined as a way to get accurate estimation. However, this is a convenience sample. Without using probability sampling, you have no way of determining the degree to which the sample is “accurate” or representative. The lines from 71-74 do not make sense with this kind of sampling.

2.More information on how the people were contacted for the survey is needed. If the questionnaire was sent to various places, how did they know that they would reach at least 700? The whole sampling and administrative procedure needs more explanation.”

Respons 3:Specific instructions for convenient sampling have been added.(line 79-83 )

“The inclusion criteria for the study population were HCPs involved in the governmental free HPV vaccination program in Shenzhen who voluntarily participated in the survey. The exclusion criterion was refusal to participate in the questionnaire. The survey for this study was collected by sending electronic questionnaires to HCPs who attended the training for the free HPV vaccination program in Shenzhen.”

Point 4:results

  1. How was a “validity rate” determined? What is meant by this? (line 114)

  1. In running the chi-square analyses (table 1), were there any issues with small expected cell sizes? Some of the cell frequency values are quite small which likely leads to problems with the chi-square test.

     3.Table 2—I would indicate i the table title that this is the N and % of those getting the answers correct. This is not clear as it is currently formatted.

      4. I’m curious as to why the authors did not look at the socio-demographic factors as possible predictors for whether an HCP has recommended the HPV vaccine. Are there too few people in the “no” category for meaningful analysis? (This then also highlights the point I make above about the potential problem with chi-square analysis.)

Respons 4:

  1. validity rate 93.0% means that a total of 828 questionnaires were received in this study, and 770 of them were able to be analyzed.

  1. The chi-square analysis of Table 1 was strictly conducted in accordance with the application conditions of the R*C table, which are as follows: 1) the number of cells with theoretical values less than 5 in the R*C table cannot exceed 1/5; 2) there cannot be theoretical values less than 1.

  1. It has been explained in the variable description of Table 2 that the values in the table represent the number of correct responses.In the first row of the first column of Table 2(Question/Correct)

  1. Yes, we did not analyze the association between sociodemographic factors and the recommendation of HPV vaccination because the number of participants in the "No" category was relatively small. However, following your suggestion, we added an analysis of the differences in HPV and HPV vaccine knowledge scores between those who recommended and those who did not recommend the HPV vaccine.Lin4175-187

“In addition, the Mann-Whitney U test was used to analyze the differences in HPV and HPV vaccine knowledge scores between healthcare professionals who had recommended the HPV vaccine and those who had not recommended it. The results showed that the distribution of HPV and HPV vaccine knowledge scores between the two groups of healthcare professionals was inconsistent. The average knowledge score of healthcare professionals who had recommended the HPV vaccine was 12.09 ± 2.02, while that of healthcare professionals who had not recommended the HPV vaccine was 10.76 ± 2.67. The average rank of knowledge scores for healthcare professionals who had recommended the HPV vaccine was 392.15, while that of healthcare professionals who had not recommended the HPV vaccine was 267.30. The Mann-Whitney U test results indicated that there was a statistically significant difference in HPV and HPV vaccine knowledge scores between healthcare professionals who had recommended the HPV vaccine and those who had not recommended it (U=10098.500, P<0.001).”

Point 5:discussion

 It is interesting to see that in China, it is the private practices that have the lowest quality and knowledge. This is very different from the U.S. context where the private practices would have the highest quality, knowledge, and pay.

Respons 5: This may be due to the differences in healthcare systems and medical service markets between China and the United States. In China, public hospitals are the main healthcare service providers, funded and managed by the government. Private clinics are usually smaller in scale, with chaotic management and limited resources such as doctors' expertise and medical equipment. Therefore, overall, their quality and knowledge level are relatively low. In addition, there is fierce price competition in the Chinese healthcare market, and private clinics charge relatively low fees, resulting in lower compensation for doctors.

Reviewer 3 Report

Dear authors, the work is more than justified and well written. Just pointing out that the use of such long results tables creates confusion and makes the reader get lost, it is my suggestion that you review them and adapt them to a smaller format.

Author Response

Point 1:

Just pointing out that the use of such long results tables creates confusion and makes the reader get lost, it is my suggestion that you review them and adapt them to a smaller format.

Respons 1:The size of the table has been adjusted to be smaller.

Reviewer 4 Report

See attached file

Author Response

Point 1:abstract

Please introduce CHC acronym before using it in the text.

Respons 1:Added "community health center" description before CHC. Line23

Point 2:introduction

For the cervical cancer is better used “mortality” than “fatality” (line 32)

Respons 2:Changed “fatality” to “mortality” line32

Point 3:results

1.The information reported in lines 115-117 is not correct.

" The majority of respondents (38.8%, n=299) were between the ages of 31 and 40. The female

population was extremely 714 high (92.7%)". Please correct it.

2.Please, in table 1 insert a column with total.

3.In table 2, I don’t find the statistical significance that you write in the lines 129-132. Please report it in table 2

4.In table 3, Note that HCP with higher income is not significant for knowledge of HPV and HPV

vaccine. See lines 142-143.

5.In table 4 “HPV Vaccine Recommendation behavior” in addition to the reasons please reported if there are differences between those who recommend and those do not recommend for HPV knowledge.

6.Please, in table 1 and in table 3 insert “HCP” after other in the variables “MAJOR” and “Type of license”.

7.Please evaluate if the differences of level of knowledge of HCP have a role in recommendation

vaccine. Are there differences between those who recommend and those do not recommend for

HPV knowledge? Is there any association between disinformation or specific area of disinformation and the recommendation HPV vaccine (Yes/No)?! Please investigate the determinants of disparity in recommendation.

Respons 3:

  1. The content has been modified to “The majority of respondents in Level I/Regional CHC(42.3%, n=115),Level II CHC(41.6%, n=128) and Private CHC(22.0%, n=20) were between the ages of 31 and 40. The female respondents' population was extremely high (94.1%).”line129-131

  1. added a column with “total” in table 1

  1. Explanation about significance is provided below Table 2 in the article.(line 192-196)

“Kruskal-Wallis H:H=28.441,P<0.001. Kruskal-Wallis 1-way ANOVA(k samples):Private CHC-Level I/Regional CHC( adjsted P<0.001); Private CHC-Level II CHC ( adjsted P<0.001);Private CHC-District Hospitals( adjsted P<0.001); Level I/Regional CHC-Level II CHC( adjsted P=1.000); Level I/Regional CHC-District Hospitals( adjsted P=1.000); Level II CHC-District Hospitals( adjsted P=1.000).”

  1. After-tax annual income showed significant differences, with a P value of 0.029 in the multiple logistic regression analysis.Table 3

  1. Further analysis was conducted in the report to examine the differences in knowledge levels of HPV and HPV vaccines between recommenders and non-recommenders.The result is that there is a difference between the two.The knowledge scores of HPV and HPV vaccine were higher among recommenders than non-recommenders.line175-187

“In addition, the Mann-Whitney U test was used to analyze the differences in HPV and HPV vaccine knowledge scores between healthcare professionals who had recommended the HPV vaccine and those who had not recommended it. The results showed that the distribution of HPV and HPV vaccine knowledge scores between the two groups of healthcare professionals was inconsistent. The average knowledge score of healthcare professionals who had recommended the HPV vaccine was 12.09 ± 2.02, while that of healthcare professionals who had not recommended the HPV vaccine was 10.76 ± 2.67. The average rank of knowledge scores for healthcare professionals who had recommended the HPV vaccine was 392.15, while that of healthcare professionals who had not recommended the HPV vaccine was 267.30. The Mann-Whitney U test results indicated that there was a statistically significant difference in HPV and HPV vaccine knowledge scores between healthcare professionals who had recommended the HPV vaccine and those who had not recommended it (U=10098.500, P<0.001).”

  1. Added HCP after other in the variables "major" and "type of license" in Table 1 and Table 3.

  1. According to the current research data, it is not possible to analyze and determine the factors that contribute to the difference in HPV and HPV vaccine knowledge between recommenders and non-recommenders, although the data indicates that such a difference exists.

Respons 4:discussion

  1. Please, Stress the role of health professionals in vaccination improvement.

         2.To improve vaccination is could be increased HCP's knowledge, changed mode of communication HCP-patient, increased HCP awareness, .......Which is the better?

Respons 4:

  1. At the end of the discussion, the importance of increasing the HPV vaccination rate among healthcare providers and reducing the incidence of HPV-related diseases was emphasized.

         2.Because there is no analysis or comparison of the actual impact of HCP knowledge and recommendation on HPV vaccination in this study, there is not enough evidence to determine which one is better.

Round 2

Reviewer 1 Report

In the revised version of the manuscript (ID: vaccines-2359278), significant corrections were made, which contributed to the text's logical flow and better clarity of the presented results. The authors took into account almost all my comments and suggestions. A discussion of the study's strengths and limitations is now included in the Discussion section. I thank the authors for their efforts in revising the paper.  

The quality of English language is appropriate.   

Author Response

Thank you for your feedback and for taking the time to review our revised manuscript. We greatly appreciate your valuable comments and suggestions, which have helped to improve the clarity and overall quality of our work. Thank you again for your support and feedback.

Reviewer 2 Report

The authors have done a fair job in addressing the earlier review comments. However, there are some issues that still need to be addressed. The following edits are strongly suggested:

* Delete lines 77-80, from "To ensure accurate..." through to "of 63% to 86%)." This kind of probabilistic/statistical reasoning does not make sense given the type of sampling used. The information is also not necessary.

* The added information about the use of Mann-Whitney to test differences in knowledge scores (lines 95-98) should be moved to the paragraph on statistical analysis. It does not fit under data collection/questionnaire information.

* Delete the sentence on lines 114-116 that begins with "To investigate..." and ends with "were used." This repeats the sentence that follows it.

* Delete reference to a validity rate on line 129. This is an incorrect use of the term validity. Either delete the parenthetical remark on line 129 or replace the term "validity" with the term "usability" since that is what the authors say they mean.

* Delete the words "and representative" from line 335. Given convenience sampling we cannot know that this survey is representative.

Extensive editing is needed

Author Response

Thank you for your feedback. We appreciate your acknowledgment of our efforts to address the previous review comments. We will carefully consider your suggestions and make the necessary edits to address the remaining issues.

Point 1:Delete lines 77-80, from "To ensure accurate..." through to "of 63% to 86%)." This kind of probabilistic/statistical reasoning does not make sense given the type of sampling used. The information is also not necessary.

Response 1This text has already been deleted.

To ensure accurate estimation of HCPs' awareness rate of HPV,a target sample size of 700 was predetermined.This sample size was determined based on the ability to estimate the awareness rate with 95% precision, given a previously observed rate of 74% (with a 95% confidence interval of 63% to 86%).

Point 2:The added information about the use of Mann-Whitney to test differences in knowledge scores (lines 95-98) should be moved to the paragraph on statistical analysis. It does not fit under data collection/questionnaire information.

Response 2The additional information about using the Mann-Whitney test to assess differences in knowledge scores has been moved to the statistical analysis paragraph.(lines 110-113)

Point 3:Delete the sentence on lines 114-116 that begins with "To investigate..." and ends with "were used." This repeats the sentence that follows it.

Response 3"To investigate the characteristics associated with levels of HPV vaccine knowledge, dichotomous logistic regression and multi-variable logistic regression were used."has already been deleted.

Point 4:Delete reference to a validity rate on line 129. This is an incorrect use of the term validity. Either delete the parenthetical remark on line 129 or replace the term "validity" with the term "usability" since that is what the authors say they mean.

Response 4The term "validity" has been replaced with "usability".

Point 5:Delete the words "and representative" from line 335. Given convenience sampling we cannot know that this survey is representative.

Response 5"and representative" has been removed
